# Effect of Sorbent Additives to Copper-Contaminated Soils on Seed Germination and Early Growth of Grass Seedlings

**DOI:** 10.3390/molecules26185449

**Published:** 2021-09-07

**Authors:** Katarzyna Możdżeń, Beata Barabasz-Krasny, Tatiana Kviatková, Peiman Zandi, Ingrid Turisová

**Affiliations:** 1Institute of Biology, Pedagogical University of Krakow, Podchorążych 2 St., 30-084 Kraków, Poland; beata.barabasz-krasny@up.krakow.pl; 2Department of Biology and Ecology, Faculty of Natural Sciences, Matej Bel University in Banská Bystrica, Tajovského 40, 97401 Banská Bystrica, Slovakia; tatiana.kviatkova@umb.sk (T.K.); ingrid.turisova@umb.sk (I.T.); 3International Faculty of Applied Technology, Yibin University, Yibin 644000, China; z_rice_b@yahoo.com

**Keywords:** copper soils, germination, grasses, natural sorbents, potentially toxic elements, recultivation

## Abstract

Heavy metal and metalloid-contaminated soil is a serious barrier to colonization for many plant species. The problem of the elimination of toxic waste accumulated in technogenous soils in many highly transformed regions is extremely important. Hence, another attempt was made to analyze the effect of the addition of sorbents (BCH—biochar, B—bentonite, ChM—chicken manure, OS—organo-zeolitic substrate) to contaminated copper soil on the germination and early growth of Eurasian common grass species (*Agrostis capillaris*, *A. stolonifera*, *Festuca rubra* and *Poa pratensis*), which could potentially be used in recultivation. This experiment was based on the laboratory sandwich method. Standard germination indexes, morphometry and biomass analysis were used. The percentage of germinating seeds was lower in each of the soil variants and sorbents used compared to the control. Dry mass was positively stimulated by all sorbents. The response to the addition of sorbents, expressed as the electrolyte leakage of seedlings, was different depending on the species and type of sorbent. Among all sorbents, the most positive effects on germination and growth were observed in the case of OS. Overall, the response to the addition of sorbents was different in the studied species, depending on their stage of development.

## 1. Introduction

Potentially toxic elements (PTEs) occur naturally in the environment. However, weathering and volcanic eruptions and other natural activities produce only insignificant concentrations of them. The most important sources of high contents of PTEs, above all in soil, are anthropogenic activities such as mining and smelting operations, industrial production and urbanization [1,2,3,4].

Soils contaminated with PTEs have sporadic to absent vegetation cover and are exposed to wind, water erosion, solar radiation and solifluction; therefore, PTEs can spread to the surrounding uncontaminated soil [5,6,7]. Increased concentrations of PTEs in the soil currently represent one of the most serious environmental problems, as they adversely affect the quality of the environment and are toxic to living organisms [3,8,9]. To eliminate their harmful effects, it is necessary to reduce their content in the soil by remediation methods. In the past, conventional methods have been used such as soil washing, soil replacement, landfilling, etc. The disadvantages of these techniques are impracticality, their time-consuming nature, money and material requirements [10,11,12].

Currently, there is a trend to use acceptable forms of remediation, not only from an ecological and economic point of view [11], but also from a social point of view. The use of sorbents, especially natural ones, appears to be one of these trends. Sorbents are solid substances that are able to adsorb or absorb another substance [13]. Frequently used sorbents include, for example, bentonite, perlite, biochar, CaCO_3_ and manure [14].

Bentonite is one of the most abundant minerals in soil [15]. It is a kind of clay rock composed primarily of montmorillonite; it mainly includes minerals from the group of smectite clays, quartz impurities and minerals containing kaolinite and illite [16]. It has been shown to be an effective sorbent for Pb, Cd, Cu, Zn [17,18,19], Cr, Hg [20], Ni and Mn from aqua media [21], improve water use efficiency, increase soil moisture, and increase the availability of water, organic carbon and potassium in the soil, which has a positive effect on the emergence and yield of plants [22,23].

Biochar is a solid carbonized product of the thermal decomposition of organic matter with high aromaticity and high carbon content at a temperature below 900 °C under conditions of oxygen deficit [24,25]. Plant-based biochar is usually poor in nutrients, while manure is rich in nutrients and is treated as a fertilizer [26,27]. This sorbent increases water retention [28], specific surface area, cation exchange capacity, total N, organic C and soil nutrient availability [29,30] and supports carbon sequestration and greenhouse gas mitigation [31]. It can immobilize heavy metals by absorbing them on its surface, increase soil pH and induce a liming effect, effectively reducing the mobility of As, Cd, Cu, Ni, Pb and Zn [32,33]. It has a positive effect on plant growth and microbial activity in soils contaminated with metals [27,34]. Biochar can significantly reduce the content of Pb, Zn, Cd and Cu [35,36,37].

Organo-zeolitic substrates consist of a mixture of sorbents of biological origin (e.g., manure, biochar, peat, etc.) and an alkaline zeolite, and optionally also another sorbent of abiotic origin (e.g., CaCO_3_). Each of these components has shown a positive effect on reducing the content of some PTEs, e.g., perlite: Pb, Cu, Ni, Cd, [38]; chicken manure: Cd, Ni [39,40]; CaCO_3_: Pb, Cu, Cd [41,42]. Zeolites have a positive effect on the fertility and sorption properties of the soil and the formation of phytomass [43,44]. Acting as a free soluble fertilizer, they improve the water balance and sorption properties, especially in light, sandy soils, which translates into higher yield and better quality. Soil fertilized or enriched with mixtures of organic zeolites regulates the release of N and minimizes the formation of reactive N(NO_3_^–^) [45].

The availability of Cu and other metals in soils is influenced by the content of organic matter, clay and soil pH [46]. In general, a high proportion of organic matter may reduce the bioavailability of heavy metals [47]. A large specific surface of clay minerals and an increase in soil pH contribute to a greater immobilization of heavy metals, especially those in the form of cations [48,49,50,51]. The aim of this experiment was to investigate the effect of the additives of five selected sorbents on the germination and growth of Eurasian common grasses, on a substrate containing soil contaminated with copper and other potentially toxic ingredients. These studies are a continuation of the experiment described in the study by Turisová et al. [52].

## 2. Results

### 2.1. Germination Capacity

Regardless of the type of grass, the highest germination capacity was demonstrated for seeds grown in the control group on distilled water (Table 1 and Table 2). The germination rate calculated after 3 days of germination (GR3) for *Agrostis capillaris* and *A. stolonifera* was similar between the control and the copper-contaminated soil without sorbents (Cu). In *A. capillaris*, the sorbents used caused a decrease in GR3 values compared to the control and the Cu-contaminated soil (Cu). No significant differences in the amount of germinated seeds were also observed between seeds germinating on Petri dishes with Cu alone and with the addition of organic substrates (OS). The seeds of *A. stolonifera* after 3 days of treatment with sorbents germinated in a lower amount than the seeds germinated in the control sample. At that time, on the Petri dishes with the addition of biochar (BCH), no germinated seeds were observed. The GR3 index of *Festuca rubra* seeds was significantly the highest in the sample with distilled water, and the lowest in the chicken manure sample (ChM). *Poa pratensis* showed the highest percentage of germinated seeds on a substrate with Cu alone compared to all other samples.

Germination rates after 7 days of germination (GR7) for *A. capillaris*, *A. stolonifera* and *F. rubra* were similar in all soil types and in the control. In the case of *P. pratensis*, similar GR7 values between the control and Cu and with the addition of OS were observed. In the remaining cases, the number of germinated seeds was significantly lower than in distilled water (control).

The values of grass germination expressed by the speed of emergence (SE) were the highest in the control. In *A. capillaris*, the significantly lowest SE values for seeds grown with the addition of ChM, bentonite (B) and BCH were found. *A. stolonifera* seeds achieved the lowest values of the SE index on BCH, ChM and OS. In *F. rubra*, each of the sorbents decreased the SE values compared to the control. The lowest values of this index were observed for seeds germinated on ChM. The SE index in *P*. *pratensis* was similar between the control and Cu. The other sorbents significantly decreased the SE values in relation to the seeds from the control sample.

The germination index (GI) in *A. capillaris* was similar between the control and Cu, and significantly lower in the presence of sorbents. For *A. stolonifera* seeds, no statistical differences were found in the value of this index. The GI of *F. rubra* was significantly lower in the presence of each of the sorbents and Cu. In the case of *P. pratensis*, the highest GI values were observed for Cu in relation to the control and sorbents. 

The mean germination rate (MGT) for germinating seeds of *A. capillaris*, *A. stolonifera* and *Festuca rubra* was similar and did not differ between the control and the sorbents. In the case of *P. pratensis*, a decrease in its value was observed on BCH, B and ChM media compared to the control (Figure 1A).

The seed vigor index (SVI) for *A. capillaris* was similar between the control and ChM and OS. A significant reduction in the SVI value was demonstrated for the remaining sorbents in relation to the control sample. The germinating seeds of *A. stolonifera* achieved the lowest values of the SVI index on the substrate with Cu, both in relation to the control and other tested sorbents. Significantly lower SVI values were observed on the BCH and B media. In the case of *F. rubra*, SVI was the lowest for seeds germinated on Cu. In *P. pratensis*, significant differences were found between the seeds germinated on the B sample compared to other sorbents and the control (Figure 1B).

The values of the CRG index in *A. capillaris* were significantly the highest for seeds germinated on ChM compared to the control and other sorbents. In *A. stolonifera*, the CRG achieved the lowest values on BCH, B and ChM. For seeds of *F. rubra*, no changes in the values of this parameter were found. In the case of *P. pratensis*, significant differences between OS and the control and Cu were observed (Figure 1C).

The contamination soil effect response index (RI) expressed in % of control showed that the germination of *A. capillaris* seeds was inhibited on Petri dishes with Cu, BCH and OS. In *A. stolonifera*, a negative effect on the germination capacity of Cu, BCH and B was found. In the case of *F. rubra* seeds, BCH and B limited seed germination. In *P. pratensis*, it was shown that all sorbents inhibited germination. Only soils with Cu alone, devoid of sorbents, had a positive effect (Figure 1D and Table 2).

### 2.2. Biometric Analysis

Biometric analysis of whole seedlings and roots of *Agrostis capillaris* showed similar values between the control and OS (Figure 2A, Figure 3A and Table 2). For seeds treated with Cu and B, the significantly lowest elongation growth was observed. The aboveground part of *A. capillaris* seedlings was the longest on the OS compared to the control and other sorbents. These results were confirmed by calculating the growth inhibition index (IP) of the seedlings expressed in % of control. The IP values for all examined organs of *A. capillaris* seedlings were generally positive, which indicated a negative impact of soil modification on their elongation growth. The exception was the addition of OS sorbent, which slightly stimulated the growth of whole seedlings (negative values in the figure) (Figure 2E and Table 2).

The length of the whole seedlings of *A. stolonifera* was significantly the shortest in the seedlings grown on Cu in relation to the control and all sorbents (Figure 2B and Figure 3B). The root was most inhibited by Cu soil and Cu soil with bentonite (B) compared to the control (distilled water) and other sorbents. The aboveground part was significantly inhibited only by Cu soil. The obtained results were confirmed by measuring the IP of *A. stolonifera* underground and aboveground organs (Figure 2F). The soil contaminated with Cu inhibited the elongation growth of all seedling organs. Similarly, Cu soils with B addition negatively influenced the growth of roots and whole seedlings of *A. stolonifera* (positive values in the figure). The remaining sorbents showed a positive effect on seedling elongation (negative values in the figure).

The morphology of the seedlings of *F. rubra* showed that soils contaminated with Cu alone and those with sorbents—B and ChM—inhibited the growth of whole seedlings (Figure 2C and Figure 3C). In the case of underground parts, the significantly highest increase was observed for the roots of seedlings germinated on BCH, and the lowest on Cu, compared to the control. The aboveground part was significantly shortest in seedlings grown on Cu compared to the remaining samples. Positive values of IP confirmed the negative effect of Cu, B and ChM on the growth of *F. rubra* seedlings, and the negative values of the positive effect of BCH and OS (Figure 2G).

The length of whole *P. pratensis* seedlings did not differ between the control and OS (Figure 2D and Figure 3D). In the remaining cases, growth inhibition was observed, as compared to the control. The shortest roots were found for *P. pratensis* seedlings grown on substrates with only Cu and ChM in relation to the remaining studied groups. The aboveground part of *P. pratensis* was the longest in the control and OS. In the remaining cases, the growth of this organ was inhibited.

The values of the IP index in most of the samples were positive for all studied organs of *P. pratensis* seedlings (Figure 2H). The exceptions were the aboveground parts of seedlings germinated on BCH, B and OS and the length of the whole seedlings germinated on OS, which reached negative values, which indicated a positive effect of sorbents on their elongation.

### 2.3. Fresh and Dry Masses and Total Water Content

The fresh mass of single seedlings of *Agrostis capillaris* was smaller for seedlings grown on B and ChM compared to the control and other sorbents. *A. stolonifera* seedlings achieved similar mass values only on the Petri dishes with distilled water and OS. In other cases, the fresh mass decreased. The values of this parameter for *Festuca rubra* significantly decreased in seedlings germinated on Cu and B compared to the control and other sorbents. The fresh mass of *Poa pratensis* seedlings was smaller in each treatment, relative to the control (Figure 4A).

The dry mass of *A. capillaris* seedlings was the largest for seedlings grown on Petri dishes with Cu compared to the control and sorbents. In *A. stolonifera* and *P. pratensis*, no statistically significant differences were found in the dry mass values. Compared to the control, each of the sorbents caused an increase in dry mass of *F. rubra* seedlings. The largest increase in the value of this parameter was observed for seedlings on BCH (Figure 4B and Table 2).

The percentage of dry mass of *A. capillaris* seedlings did not differ significantly between the control and the Cu and sorbents. The highest percentages of dry mass of *A. stolonifera* seedlings were found on Petri dishes with B compared to the control and other sorbents. For *F. rubra* and *P. pratensis*, the values of this parameter were the lowest in the control (Figure 4C).

Total water content (TWC) in *A. capillaris* achieved significantly the highest values in the control in relation to Cu, B and OS. *A. stolonifera* seedlings showed no significant differences in the percentage of water content. Compared to the control, each of the sorbents caused a significant reduction in TWC in *F. rubra* and *P. pratensis* seedlings (Figure 4D).

### 2.4. Electrolyte Leakage

For *Agrostis capillaris* seedlings, significant differences were found only between the control and OS. The remaining sorbents increased the percentage of electrolyte leakage, but these differences were not statistically significant. In *A. stolonifera*, a significant increase in electrolyte leakage was noted for B, ChM and OS compared to the control. In *Festuca rubra*, BCH significantly reduced the destabilization of cell membranes compared to the distilled water, other sorbents and the soil itself contaminated with Cu. In *Poa pratensis*, each of the sorbents increased the flow of electrolytes through the membranes. For seedlings grown on Cu and B, statistically significant differences were observed (Figure 5 and Table 2).

## 3. Discussion

One of the toxic anthropogenic factors is heavy metals and metalloids present in water and soil. The potential toxicity of PTEs in soil depends on their specific form, reactivity, mobility, concentration and availability to living organisms. The bioavailability of metals in the soil is constantly changing and depends on various physicochemical, biological and environmental parameters [53]. At low concentrations, heavy metals are essential and useful micronutrients, while at high concentrations they are highly harmful to microorganisms, plants and animals [54].

Seed germination is one of the most important stages in plant life. This process is sensitive to the chemical and physical conditions in the rhizosphere [55]. Generally, heavy metals inhibit seed germination and seedling development by the inhibition of storage food mobilization, a reduction in radical formation, the disruption of cellular osmoregulation and the degradation of proteolytic activities [56]. For example, an excess of Cu, according to many researchers, leads to a reduction in seed germination [57,58,59,60]. In germinating seeds, Cu interferes with the release of nitrogen from the storage tissue, affecting not only the development of the seed embryo, but also the overall activity of proteases [61]. It induces the mobilization of biomass by the release of glucose and fructose, thus inhibiting the breakdown of starch and sucrose. The toxicity of Cu affects the overall metabolism of the plant, water uptake, the lack of mobilization of material reserves, etc. [62]. This element moves slowly in the soil, usually as an organic complex, and accumulates in the soil, mainly on its surface [63].

The influence of metals on seed germination also depends on the interspecific differences in the structure of their seed coat and the content of enzymes produced in the aleurone layer [64]. These mechanisms probably contributed to the differences in the response to Cu observed in the experiment performed here. Among the studied species of grasses, the seeds of *Poa pratensis* were the most resistant to the toxic effect of copper and germinated under the conditions of an increased concentration of this element in the soil (Table 2).

In the other grass species, the germination process was clearly inhibited. Although the seed coat acts as a major barrier to the harmful effects of heavy metals, most seeds and seedlings show a reduction in germination and vigor in response to heavy metal stress [65]. The seed coat has a wide range of anatomic forms that exist in no other plant organ or tissue; it is also the main barrier to metals and prevents the contamination of embryos until it is burst by the germinating embryonic root [66].

In the experiment carried out here, it was noted that the addition of sorbents did not have a positive effect on the germination capacity of seeds in most cases (Figure 1 and Table 1). It was found that the addition of organo-zeolites (OS), and to a lesser extent chicken manure (ChM), had the most beneficial effect on the germination parameters. First of all, it was the organo-zeolitic substrate, regardless of the type of grass, in most of the calculated indexes that showed a stimulating effect on the discussed parameters (Table 2). Its positive effect can probably be partly explained by the fact that it improves the efficiency of water and plant nutrient use and increases the water absorption of the substrate [45]. It should be emphasized that the response to sorbents was specific and depended on the species of grass and type of sorbent (Figure 1).

Compared to the control, the sorbents also did not show a clear effect on the growth of seedlings (Figure 2 and Figure 3). In the summary of elongation growth parameters, it can be concluded that only OS in all tested grass species positively influenced the development of seedlings. On the other hand, the mass analysis showed a positive effect of sorbents on the accumulation of dry mass of seedlings (Figure 4). The increase in biomass can be attributed to the presence of sorbents, which most likely influenced the immobilization of Cu and the increased accumulation of nutrients through their slow release [45,67,68]. The percentage of electrolyte leakage, regardless of the seedling species, was higher than in the control and similar to the EL value of seedlings grown on the soil contaminated with Cu, without the addition of sorbents (Figure 5). It can be assumed that the applied experiment model using the sandwich method had a significant impact on the obtaining of such results. Such reactions of seedlings most likely resulted from the too short contact time between the seeds and the soils, which made it difficult to record a clear response to the applied remediation substances, contrary to the reaction of the same seeds treated with aqueous extracts from soil contaminated with copper and its modification with sorbents. Perhaps direct contact with chemical compounds extracted from the soil allowed us to verify to a greater extent which of the sorbents positively influences the germination and early growth of grass seeds [52].

In many cases, at the stage of seed germination and early grass growth, the tested sorbents did not show effective activity in copper immobilization. The addition of organic sorbents often even limited seed germination as compared to the control (distilled water). It should be noted, however, that the organo-zeolitic substrate (OS) turned out to be the most effective substance in soil reclamation with Cu, both in the germination and growth stages. However, it cannot be considered a universal sorbent as its effect was not positive in all cases. According to Finch-Savage and Bassel [69], there is no clear description of the soil conditions that lead to good or poor plant emergence. This is because the seedlings are not at all sensitive to the type or condition of the soil, but are extremely sensitive to the physical stresses that the soil exerts during seed germination and seedling growth. Therefore, it is important to understand the physical nature of the soil, especially anthropogenically transformed soil [70]. With the right selection of species for sowing, sorbent selection procedure and the right time of their use, sorbents can be utilized in the reclamation of copper-contaminated sites, where they will positively affect the development of the vegetation cover [65,70].

## 4. Materials and Methods

### 4.1. Study Area

The study material was the copper-polluted soil obtained from the Maximilián heap (48°48′29″N 19°07′59″E), located in the village of Špania Dolina (near Banská Bystrica city, Slovakia) (Figure 6). The average copper content in the soil samples was 1099 mg·kg^−1^, which is several times higher than the EU limit value according to Council Directive 86/278/EEC (50 to 140 mg·kg^−1^) [71]. The characteristics of this area have been described in detail in the paper of Turisová et al. [52]. In these areas, copper and silver ore mining was carried out over several centuries. The area of Špania Dolina ore field represents one of the historically most distinguished copper mining deposits in Europe. Here, the mineralization forms a 4 km long and 1.5 km wide vein stretching between the hill Panský diel (48°47′53.66″N 19°08′57.95″E) on the south and the village Staré Hory on the north to the east of Starohorský potok [72]. The first written reports of ore mining in the area of Špania Dolina are from the 11th century (from the year 1006). At first, copper and silver ore was mined in Staré Hory (on historical deposit Haliar). Later, the mining expanded further south from already the extinct village Richtárová and Špania Dolina locality [73]. After depleting the surface portions, the mining was gradually transferred to underground mining from the mid-14th century. The maximum development of copper ore mining with an extremely valuable silver content was in the years 1496–1546. Since the 17th century, mining gradually declined until it completely disappeared in the early 20th century [74]. Du-ring the years 1963–1964, old mine heaps (originating from the 16th–19th century) in the area of village Richtárová were re-mined by surface mining, which significantly changed the original configuration [75].

### 4.2. Soil Modification

The soil was analyzed by inductively coupled plasma mass spectrometry (ICP-MS) in Bureau Veritas Commodities Canada Ltd. in Vancouver (BC, Canada). A total of 0.25 g of samples were digested in aqua regia for ICP-MS analysis for plant and acid digestion (uses HNO_3_, HClO_4_ and HF acids heated until dryness to decompose most minerals, including silicates to metal salts, that are then back leached into concentrated HCl). Soil modifications used in the experiment are listed in Table 3 and Figure 6.

### 4.3. Short Characteristics of Sorbents

The experiment was realized with four natural sorbents: biochar, bentonite, chicken manure and organo-zeolitic substrate.

In the bibliography, there is no unambiguous legal status of biochar (BCH) that would specify whether biochar is a proper by-product or waste product [76,77]. According to the International Biochar Initiative (IBI), it is a fine-grained char with a high content of organic carbon and low susceptibility to degradation, obtained in the process of the pyrolysis of biomass and biodegradable waste. In environmental protection, biochar is used as a so-called green sorbent or a barrier component for removing pollutants [78,79]. In this study, it was bought as charcoal commercially used as charcoal for grilling. This sorbent was dried at room temperature, crushed and sieved through a 2 mm sieve [80].

Bentonite (BE) is formed from clay rocks as a result of the montmorillonitization of volcanic glass found in pyroclastic sediments such as tuffs or tuffites [81]. It is composed mainly of minerals from the smectite group (mainly montmorillonite), accompanied by the remains of pyroclastic material. Smectite group minerals are layered silicates with a packet structure. There are exchangeable cations between the packages, most often Ca^2+^, Mg^2+^ and Na^+^. Bentonite has the properties of sorbing cations and organic substances, swelling and forming thixotropic suspensions that do not sediment for a long time [82]. In this experiment, the average chemical compositions of the following substances in bentonite were: SiO_2_ 57–61%, Al_2_O_3_ 18–21%, Fe_2_O_3_ 2–3%, FeO 0.1–0.5%, TiO_2_ 0.2–0.3%, CaO 1.9–2.6%, MgO 3.0–5.0%, K_2_O 0.4–1.0%, Na_2_O 0.2–0.7% [80]. Chicken manure (ChM) is a natural fertilizer that contains large amounts of nitrogen and calcium. It is characterized by a high pH of approx. 7.5. It is a source of trace elements that can potentially accumulate in the soil with repeated use. It improves the quality and fertility of the soil. It affects the physical, biological and chemical properties of the soil [83,84]. We obtained chicken manure from domestic breeding of domestic fowl *Gallus gallus domesticus* L. [80]. Zeolites (organo-zeolitic substrate OZS) are microporous aluminosilicate minerals with a tubular structure of SiO_4_ and AlO_4_ interconnected tetrahedrons. In this study, a perlite in the form of a fine powder had an average chemical composition as follows: SiO_2_ 68.0–73.0%, Al_2_O_3_ 7.5–15.0%, Fe_2_O_3_ 1.0–2.0%, Na_2_O 1.0–2.0%, K_2_O 2.5–5.0%, CaO 0.5–2.0%, MnO max. 0.3%, TiO_2_ max. 1.0%, P_2_O_5_ max. 0.2%, MgO max. 1.0%. It is added to the soil to prevent its compaction (thus providing aeration and optimal soil moisture retention). It also supports the rooting of plants. Calcium carbonate (CaCO_3_) was pure and also in the form of a fine powder [80]. CaCO_3_ increases the pH and reduces the content of some potentially toxic elements (Pb, Zn, Cd, As), but does not affect the growth of aboveground plant biomass [85,86].

### 4.4. Seed Material with Short Botanical Description of Species

As in the previous experiment [52], in this study, four common grass species typical for Eurasian areas were selected: *Agrostis capillaris* L., *A. stolonifera* L., *Festuca rubra* L., *Poa pratensis* L. The grass seeds were bought from DLF SeeDs (Hladké Životice, Czech Republic). *A. capillaris* (common bent, browntop) is a perennial grass with very thin blades (stalks); its blades rise obliquely and are rough under the panicle. The ligule of the uppermost leaf blade is short and clipped. After flowering, the panicle disperses. This grass is common in meadows and pastures. *A. stolonifera* (creeping bentgrass, creeping bent) is a gray–green perennial grass with lying blades, rising on the tops and rooting in nodes. It has a long, jagged ligule and a panicle with a purple tinge, which is dispersed during flowering and tightly concentrated after flowering. This grass is frequently found in flooded places. *F. rubra* (red fescue, creeping red fescue) is a dense-tufted perennial grass with thin blades. It has frequently dark red and jagged leaf sheaths, and its near-ground leaves are rolled up and hair-like, and unequal in length. It has a panicle with twigs placed horizontally during flowering—the lower twig of the panicle is adjacent to the stem. This grass is common in dry meadows and pastures. *P. pratensis* (Kentucky bluegrass, blue grass) is a loose-tufted perennial grass, grey–green or yellowish in color, creating large underground stolons. The upper part of the blade is leafless. Near-ground leaves are shortened and the leaf blade of the upper leaf is shorter than its sheath. It has a pyramidal panicle with rough twigs. This grass is common in meadows and in thickets [87,88].

### 4.5. Sandwich Methods

The sandwich method was used to analyze the effect of copper-contaminated soil and sorbents added according to Fujii et al. [89]. An 0.75% agar aqueous solution was prepared from powdered agar (Chempur, Poland) and autoclaved at 121 °C for 15 min. Sterile agar was poured 5 mL into sterile Petri dishes (the first layer of "sandwiches") and allowed to solidify at room temperature. A total of 0.5 g of Cu soil or Cu soil mixed with sorbents was poured onto the solidified first surface of the agar, in the proportion given in Table 1. After the soil was evenly spread on the Petri dishes, they were poured with 5 mL of agar, which was the second layer of the "sandwich", and allowed to solidify. The control group were Petri dishes with 10 mL of agar. The experiment for each type of soil and control was carried out in 5 repetitions.

### 4.6. Seeds Preparation and Germination Conditions

Grass seeds (each separately) were sterilized in 1% acetone for 5 min, then rinsed 3 times with distilled water. Twenty-five seeds of each species were placed (with sterile tweezers) on separate sterilized, glass Petri dishes (Ø 9 cm) with agar and appropriate soil type. The control group seeds were placed on agar with distilled water. The Petri dishes with seeds were incubated in a germinator at a temperature of about 20 °C ± 2 °C, a relative humidity of about 60–70%, in the dark. After 3 and 7 days, the germinated seeds were considered those whose germinal root was equal to half the size of the seed. The experiment was performed in 5 repetitions for each of the soil types and control samples, with a total of 125 seeds of each grass species for a given soil type and control.

### 4.7. Germination Parameters

The germination capacity of grass seeds was evaluated by the germination indexes. Germination percentage—GR (global method) modified because it is calculated after 3 and 7 days of seed germination, for the determination of energy and capacity of germination, speed of emergence—SE [90], germination index—(GI) [91], mean germination time—MGT [91], seedling vigor index—SVI [92], rate of germination—RG [93], contamination soil response—RI [94].
GR (%) = [Number of germinated seeds at 3 (or 7) days × 4] × 100(1)
SE (%) = (Number of germinated seeds at the starting day of germination/Number of germinated seeds at the final days of measurement) × 100(2)
GI (a.u.) = [Number of germinated seeds/Days of first count] + [Number of germinated seeds/Days of last or final count](3)
MGT (day) = [(Number of germinated seeds in 3 days × 3) + (Number of germinated seeds in 7 days × 7)]/(Number of germinated seeds in 3 days + Number of germinated seeds in 7 days)(4)
SVI (day) = (Seedling length (cm) × Germination percent)/100(5)
RG (day) = [(n3 + n7)/((n3 × T3) + (n7 × T7)] × 100n3 and n7 = number of germinated seeds on time T3 and T7, T3 and T7 = 3 and 7 days after germination.(6)
RI (a.u.) = T/C − 1 (T < C)C is the control germination speed and T is the treatment germination speed.(7)

### 4.8. Biometric Analysis

Biometric analysis of seedlings germinated on various modifications of soil contaminated with copper was carried out by measuring the length of the underground and aboveground parts of the seedlings using the traditional method, using a caliper (Topex 31C615, Kraków, Poland) with an accuracy of 0.1 mm. Additionally, the germination inhibition index was determined using the modified formula of Mominul Islam et al. [95].
IP (%) = (1 − (L_E_/L_C_)) × 100L_E_—seedling length (cm) treated with the soil type, L_C_—seedling length (cm) treated with the distilled water (control group).(8)

### 4.9. Fresh and Dry Masses and Total Water Content

After 7 days of seed germination, fresh and dry masses were determined for single grass seedlings, on a scale (Ohaus Adventurer Pro, Morris County, NJ, USA) with an accuracy of 0.0001 g. To determine dry mass, seedlings were dried for 48 h at 105 °C in a dryer (WAMED SUP 100, Zabrze, Poland). The total water content was determined according to the formula used by Lipniak and Kliszcz [96].
H_2_O (%) = 100 − [(DM × 100)/FM] DM—dry mass, FM—fresh mass.(9)

### 4.10. Electrolyte Leakage

The percentage degree of destabilization of cell membranes was measured according to the method used by Szafraniec et al. [97]. Single grass seedlings were placed in 15 mL of distilled water in polypropylene vials and shaken for 3 h on a shaker (Labnet, Rocker, Edison, NJ, USA). After this time, each of the samples was additionally vortexed for 1 min. Using the CX-701 conductometer with an electrode (K = 1.02) (Elmetron, Zabrze, Poland), the electrolyte leakage from living cells was determined (E1). The plant material was then frozen for 24 h at temperature –75 °C for tissue maceration. One day later, the vials with the seedlings were thawed and subjected to the same shaking and measurement procedure as the live material samples to determine the total electrolyte leakage (E2). Based on the obtained data, the percentage of electrolyte leakage was calculated according to the following formula:EL (%) = (E1/E2) × 100EL—electrolyte leakage, E1—EL live seedling, E2—EL dead seedling.(10)

### 4.11. Statistical Analysis

The experiment was carried out in 5 repetitions on 125 seeds for each species of grass, each type of soil with sorbents and control (distilled water). The obtained mean results (±SD) for each of the grass species were subjected to a one-way ANOVA statistical analysis. Differences between soil modifications for each species of grass were marked with different letters in columns or rows, using Duncan’s test at *p* ≤ 0.05 in the program StatSoft, Inc. 2018 (Tulsa, OK, USA). STATISTICA (data analysis software system), version 13.1 Dell Inc., Round Rock, TX, USA.

## 5. Conclusions

The experiment carried out here showed that the response of plants to the addition of sorbents to soil contaminated with Cu depends on the plant species and its development stage. Among the four sorbents selected here, no universal one was found, the addition of which in the substrate would have a positive effect on all species analyzed here, both in the stages of germination and growth. Grass seeds germinated better on the soil contaminated with copper, rather than with the addition of any sorbents. However, the results obtained here also show that in the germination and growth stage, the addition of organo-zeolitic substrate may have a clearly positive effect, although not in all. In addition, in the studied grasses, the significantly lowest seed emergence was found for seeds grown with the addition of chicken manure, but in the later stages, this sorbent had a positive effect on the elongation growth of seedlings. Therefore, it can be concluded that other sorbents should probably be used for seed dressing and others for inducing the elongation growth of plants. Seeds treated with sorbents to accelerate germination should be available for sale, indicating what kind of contaminated soil should be sown, as well as sorbents intended for direct application during plant growth. This may be of great importance in accelerating the process of the recultivation of contaminated mining areas. However, experiments on this subject should still be conducted, as many issues are still not fully resolved here.

## Figures and Tables

**Figure 1 molecules-26-05449-f001:**
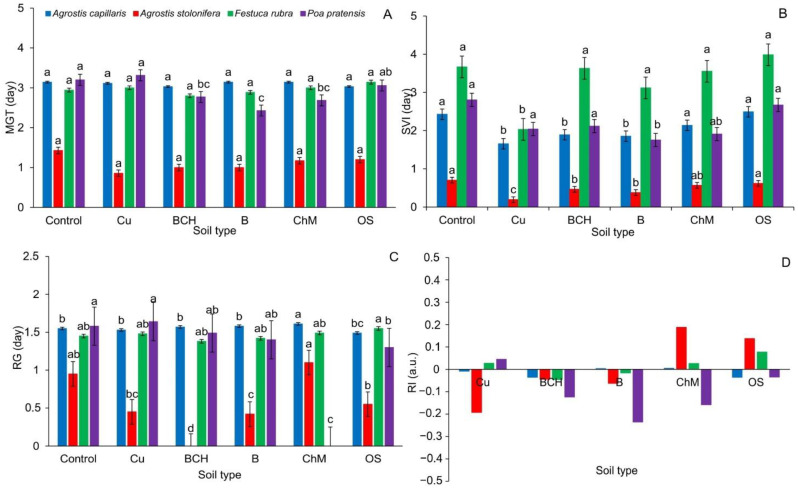
Selected germination indexes of grass seeds ((**A**)—MGT—mean germination time, (**B**)—SVI—seed vigor index, (**C**)—RG—rate of germination, (**D**)—RI—contamination soil effect response index) on various modifications of soil contaminated with copper; Cu—copper soil, copper soil with sorbents: BCH—biochar, B—bentonite, ChM—chicken manure, OS—organo-zeolitic substrate; mean values (*n* = 5; ± SD) marked with different letters (within each species) differ significantly according to Duncan’s test with *p* ≤ 0.05.

**Figure 2 molecules-26-05449-f002:**
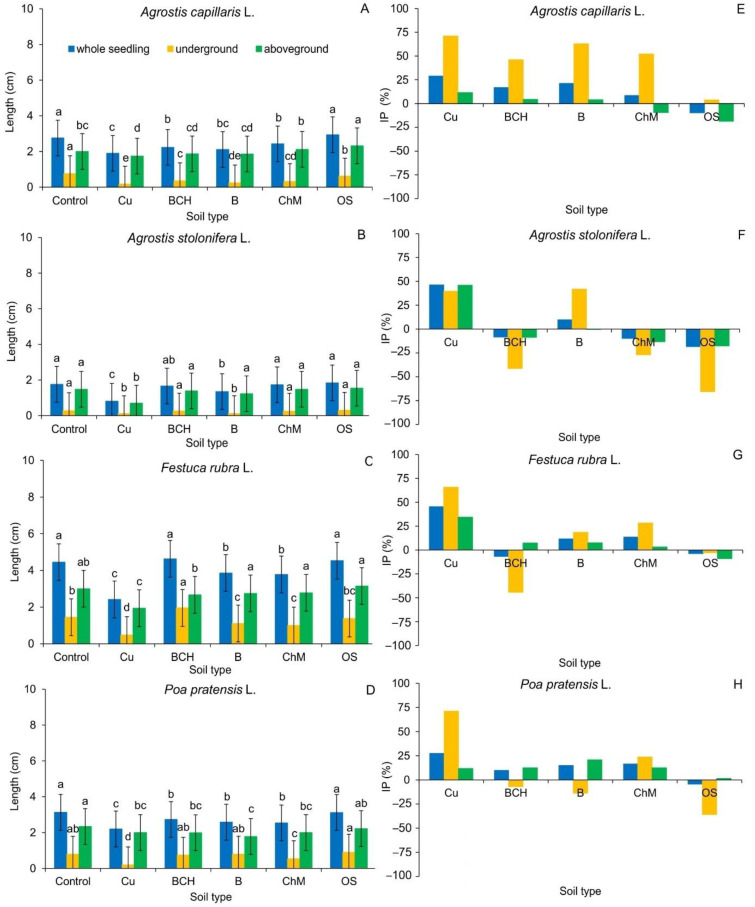
Length of grass seedlings germinated on various modifications of soil contaminated with copper; Cu—copper soil, copper soil with sorbents: BCH—biochar, B—bentonite, ChM—chicken manure, OS—organo-zeolitic substrate; mean values (*n* = 5; ± SD) expressed in (cm): (**A**)–(**D**) and in % of control: (**E**)–(**H**), marked with different letters (within each of the grass species) differ significantly according to Duncan’s test with *p* ≤ 0.05.

**Figure 3 molecules-26-05449-f003:**
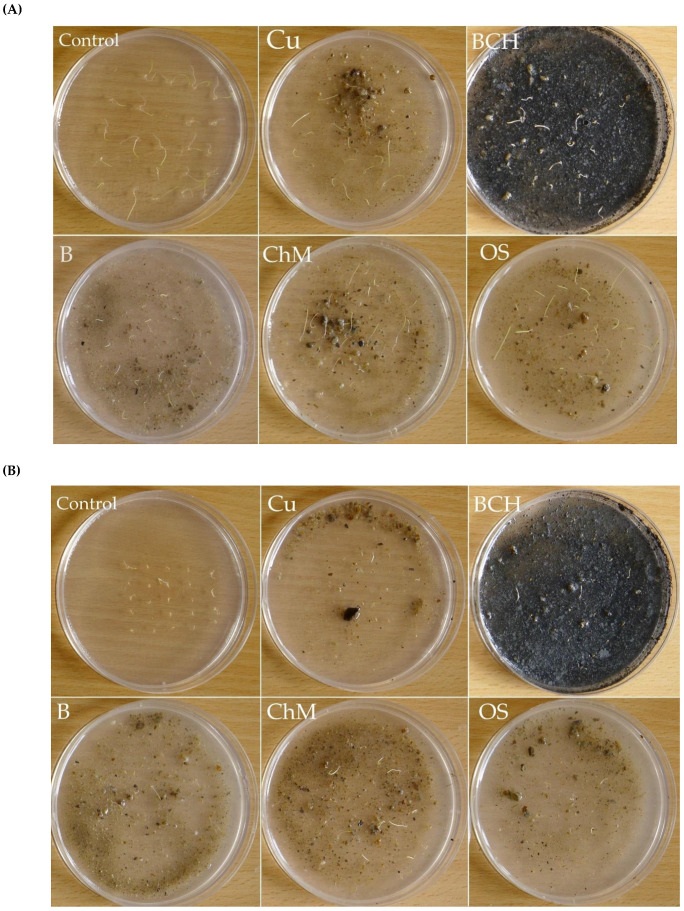
Petri dish tests after (sandwich method) 7 days of the experiment; (**A**)—*Agrostis capillaris* L., (**B**)—*A. stolonifera* L., (**C**)—*Festuca rubra* L., (**D**)—*Poa pratensis* L.; Control—distilled water, Cu—copper soil; copper soil with sorbents: BCH—biochar, B—bentonite, ChM—chicken manure, OS—organo-zeolitic substrate.

**Figure 4 molecules-26-05449-f004:**
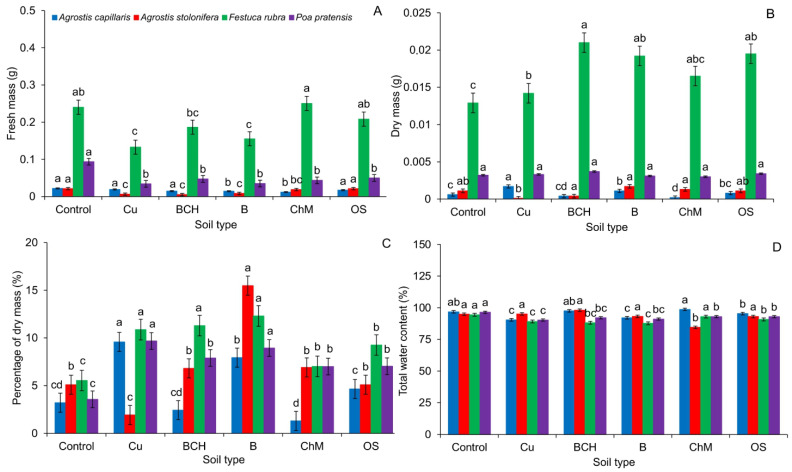
Values of fresh (**A**) and dry (**B**) mass, percentage of dry mass (**C**) and water content (**D**) in grass seedlings germinated on Petri dishes with various modifications of soil contaminated with copper; Cu—copper soil, copper soil with sorbents: BCH—biochar, B—bentonite, ChM—chicken manure, OS—organo-zeolitic substrate; mean values (*n* = 5; ± SD) marked with different letters (within each species) differ significantly according to Duncan’s test with *p* ≤ 0.05.

**Figure 5 molecules-26-05449-f005:**
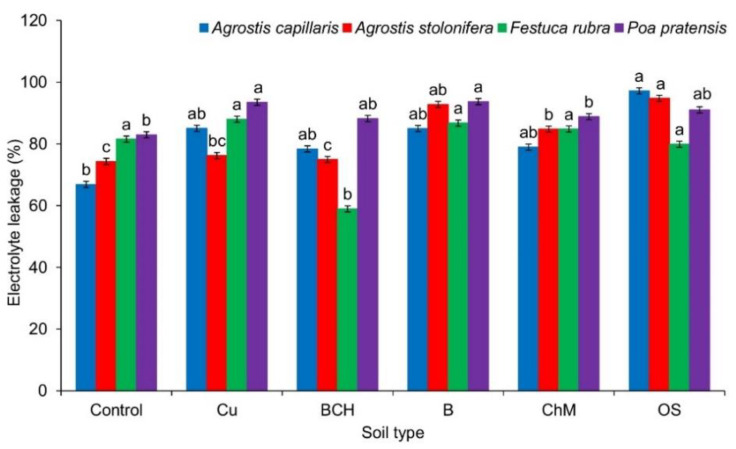
Percentage of electrolyte leakage through cell membranes of grass seedlings germinated on various modifications of soil contaminated with copper; Cu—copper soil, copper soil with sorbents: BCH—biochar, B—bentonite, ChM—chicken manure, OS—organo-zeolitic substrate; mean values (*n* = 5; ± SD) marked with different letters (within each species) differ significantly according to Duncan’s test with *p* ≤ 0.05.

**Figure 6 molecules-26-05449-f006:**
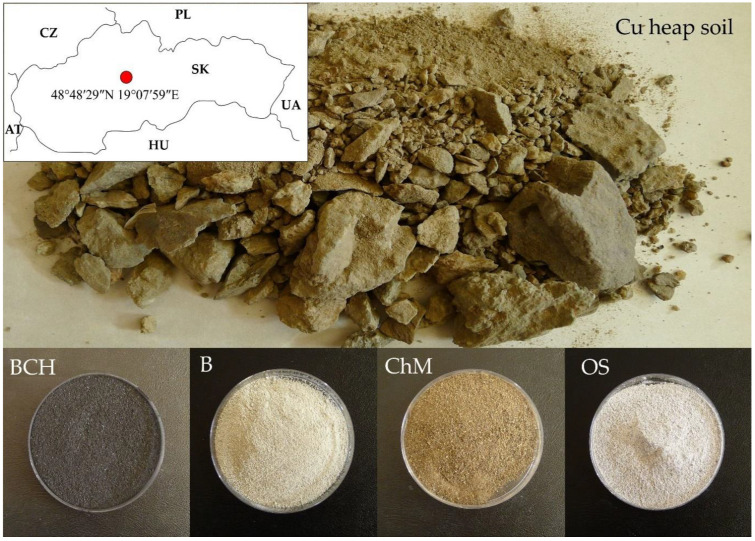
Soil types used in the experiment: Cu heap soil—copper soil; copper soil with sorbents: BCH—biochar, B—bentonite, ChM—chicken manure, OS—organo-zeolitic substrate.

**Table 1 molecules-26-05449-t001:** Values of selected germination indexes of grass seeds recorded on various modifications of soil contaminated with copper; mean values (*n* = 5; ± SD) marked with different letters (within each species, in columns) differ significantly according to Duncan’s test with *p* ≤ 0.05.

Soil Type	GR3 (%)				GR7 (%)				SE (%)				GI (a.u.)			
	AC	AS	FR	PP	AC	AS	FR	PP	AC	AS	FR	PP	AC	AS	FR	PP
**Control**	81.6 ^a^ ± 5.37	11.2 ^a^ ± 1.16	81.6 ^a^ ± 6.07	33.6 ^b^ ± 6.78	88.0 ^a^ ± 5.66	40.0 ^a^ ± 5.98	82.4 ^a^ ± 6.69	89.6 ^a^ ± 2.05	92.8 ^a^ ± 4.07	52.9 ^a^ ± 6.31	99.1 ^a^ ± 2.03	38.2 ^a^ ± 0.58	9.9 ^a^ ± 0.62	2.0 ^a^ ± 1.31	9.7 ^a^ ± 0.74	6.0 ^b^ ± 0.31
**Cu**	73.6 ^ab^ ± 7.80	7.2 ^ab^ ± 1.67	34.4 ^c^ ± 7.48	42.4 ^a^ ± 4.38	87.2 ^a^ ± 6.57	24.0 ^ab^ ± 6.02	84.0 ^a^ ± 6.34	92.8 ^a^ ± 2.42	84.5 ^ab^ ± 8.16	13.8 ^b^ ± 6.73	41.5 ^c^ ± 0.78	45.7 ^a^ ± 0.38	9.2 ^ab^ ± 0.79	1.8 ^a^ ± 0.88	5.9 ^b^ ± 0.27	6.8 ^a^ ± 0.16
**BCH**	12.0 ^c^ ± 8.94	0.0 ^d^ ± 0.00	33.6 ^bc^ ± 6.07	2.4 ^cd^ ± 6.78	84.8 ^a^ ± 8.67	28.0 ^ab^ ± 4.90	78.4 ^a^ ± 6.60	77.6 ^bc^ ± 2.72	13.4 ^c^ ± 9.39	0.0 ^e^ ± 0.00	43.4 ^bc^ ± 0.70	3.0 ^bc^ ± 0.44	4.0 ^c^ ± 1.05	1.0 ^a^ ± 0.17	5.6 ^bc^ ± 0.22	3.0 ^c^ ± 0.31
**B**	12.0 ^c^ ± 9.31	3.2 ^bc^ ± 1.94	44.0 ^b^ ± 6.46	0.8 ^cd^ ± 6.32	88.0 ^a^ ± 6.32	28.0 ^ab^ ± 4.88	80.8 ^a^ ± 6.20	68.0 ^cd^ ± 2.35	13.5 ^c^ ± 9.40	14.5 ^b^ ± 1.25	55.7 ^b^ ± 0.48	1.0 ^cd^ ± 0.35	4.1 ^c^ ± 1.02	1.3 ^a^ ± 0.32	6.6 ^b^ ± 0.44	2.5 ^c^ ± 0.23
**ChM**	5.6 ^cd^ ± 6.69	0.8 ^bc^ ± 1.85	20.8 ^d^ ± 7.48	0.0 ^d^ ± 0.00	88.0 ^a^ ± 6.52	32.8 ^ab^ ± 2.98	84.0 ^a^ ± 6.04	75.2 ^bd^ ± 2.00	6.4 ^cd^ ± 7.34	1.3 ^d^ ± 1.71	24.2 ^d^ ± 1.02	0.0 ^d^ ± 0.37	3.6 ^cd^ ± 0.81	1.2 ^a^ ± 0.57	4.7 ^c^ ± 0.27	2.7 ^c^ ± 0.37
**OS**	64.0 ^b^ ± 5.52	1.6 ^bc^ ± 1.45	39.2 ^b^ ± 4.60	0.8 ^cd^ ± 6.28	84.8 ^a^ ± 6.27	33.6 ^ab^ ± 3.36	88.0 ^a^ ± 6.40	85.6 ^ab^ ± 2.48	76.3 ^b^ ± 2.84	5.4 ^c^ ± 1.47	45.1 ^bc^ ± 0.86	1.1 ^cd^ ± 0.17	8.4 ^b^ ± 0.82	1.3 ^a^ ± 0.48	6.4 ^b^ ± 0.17	3.1 ^c^ ± 0.30

AC —*Agrostis capillaris* L., AS—*Agrostis stolonifera* L., FR—*Festuca rubra* L., PP—*Poa pratensis* L.; Cu—copper soil; copper soil with sorbents: BCH—biochar, B—bentonite, ChM—chicken manure, OS—organo-zeolitic substrate; GR3—germination rate after 3 days, GR7—germination rate after 7 days, SE—speed of emergence, GI—germination index.

**Table 2 molecules-26-05449-t002:** Comparison of the impact of soil with an increased content of Cu and with the addition of sorbents (BCH—biochar, B—bentonite, ChM—chicken manure, OS—organo-zeolitic substrate) on selected parameters of germination and growth of the studied species of grasses (AC—*Agrostis capillaris*, AS—*A. stolonifera*, FR—*Festuca rubra*, PP—*Poa pratensis*); germination indexes: RG—rate of germination, SVI—seed vigor index, RI—contamination soil effect response index; elongation growth parameters: FM—fresh mass, DM—dry mass, EL—electrolyte leakage.

Parameter	Species	Cu	Type of Sorbents			
			BCH	B	ChM	OS
Germination						
RG	AC					
	AS					
	FR					
	PP					
SVI	AC					
	AS					
	FR					
	PP					
RI	AC					
	AS					
	FR					
	PP					
**Growth of seedlings**						
Length Whole Seedling	AC					
	AS					
	FR					
	PP					
FM	AC					
	AS					
	FR					
	PP					
DM	AC					
	AS					
	FR					
	PP					
EL	AC					
	AS					
	FR					
	PP					

Note: red—inhibits, green—stimulates in relation to control.

**Table 3 molecules-26-05449-t003:** Soil modifications used in the experiment.

Abbreviation	Soil	BCH	B	ChM	OS
Full name	Cu-contaminated soil froma mining heap0.5 g	biochar	bentonite	chicken manure	organo-zeolitic substrate
Mixture proportions per 1 unit weight of soil (g)		0.2 g—biochar	0.1 g—bentonite	0.01 g—chicken manure	0.5 g—perlite0.3 g—CaCO30.1 g—chicken manure
		0.8 g—soil	0.9 g—soil	0.99 g—soil	0.91 g—soil

## Data Availability

Not applicable.

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
