# Peer review of "Effect of Sorbent Additives to Copper-Contaminated Soils on Seed Germination and Early Growth of Grass Seedlings"

_molecules, 2021, doi:10.3390/molecules26185449_

Round 1
Reviewer 1 Report
Please see in the attach.

Author Response
Reviewer 1,
We would like to thank you for review this manuscript. All suggestion have been added in the text.
Open Review
(x) I would not like to sign my review report
( ) I would like to sign my review report
English language and style
( ) Extensive editing of English language and style required
( ) Moderate English changes required
( ) English language and style are fine/minor spell check required
(x) I don't feel qualified to judge about the English language and style
Yes |
Can be improved |
Must be improved |
Not applicable |
|
Does the introduction provide sufficient background and include all relevant references? |
(x) |
( ) |
( ) |
( ) |
Is the research design appropriate? |
(x) |
( ) |
( ) |
( ) |
Are the methods adequately described? |
( ) |
( ) |
(x) |
( ) |
Are the results clearly presented? |
( ) |
(x) |
( ) |
( ) |
Are the conclusions supported by the results? |
(x) |
( ) |
( ) |
( ) |
Comments and Suggestions for Authors
Please see in the attach.
Observations regarding the paper: “Effect of Sorbent Additives to Copper-Contaminated Soils on Seed Germination and Early Growth of Grass Seedlings”, authors Katarzyna Możdżeń et al., :
Reviewer: Was the soil used in the experiment analysed in terms of copper content?
Authors: We entered information on the high copper content in line 346: "The average copper content in the soil samples was 1099 mg×kg−1, which is several times higher than the EU limit value according to Directive 86/278/EEC (50 to 140 mg×kg−1)."
Reviewer: How do you explain the assertion: “The soil was analysed by multi-acid inductively coupled plasma emission spectrometry (ICP-ES) ........”?
Authors: We better explained the method of soil analyses in lines 370 – 373: “The soil was analysed by inductively coupled plasma emission spectrometry (ICP-MS) in Bureau Veritas Commodities Canada Ltd. in Vancouver (Canada). A 0.25 g of soil was gradually heated in HNO3, HClO4 and HF to fuming and taken to dryness. The residue was dissolved in HCl.”
Reviewer: In table 3, the soil quantities treated with sorbents are different. Will this affect the action of the sorbents?
Authors: The final soil-sorbent ratio used in the experiment was determined in a pre-test (separately for each type of sorbent due to its specific effect) in order to achieve the best possible seed germination. For example, if a higher amount of chicken manure was added to the soil, the seeds did not germinate at all (they were "burned"). Therefore, we can say that the weight ratio of soil-sorbent has a significant effect on seed germination and is different for each type of sorbent.
Example from pre-test for biochar:
For bentonite (where the effect was very similar in 10% and 15% bentonite, and therefore we chose an economically more advantageous variant).
Sincerely Yours,
Authors

Reviewer 2 Report
The problem of heavy metal and metalloid contamination of soil to be a serious barrier to colonization for many plant species is very important. Therefore the effect of the addition of biochar, bentonite, chicken manure, organo-zeolitic substrate to the contaminated copper soil on germination and early growth of the grass species - Agrostis capillaris, A. stolonifera, Festuca rubra and Poa pratensis) was investigated. To this aim, the standard germination indexes, morphometry and biomass analysis were determined. In my opinion, the paper is interesting and deserves to be published after major revision.
Below I have listed some specific comments to improve the paper:
Please finish the sentence ‘effectively reducing the mobility of…’
In 4.3. Short characteristics of sorbents - some repetitions should be removed.
What was the reason to use CaCO3? Please add the characteristic of CaCO3.
As for zeolites the description ‘They are characterized by a well-developed specific surface, high sorption capacity, high ion exchange capacity and resistance to acids and high temperature’ is too general. A detailed zeolite characteristic is needed.
What does it mean ‘Its size did not need to be adjusted’?
Please explain why the grass seeds germinated better on the soil contaminated with copper?
In Conclusion, significant advantages of the proposed solution should be mentioned.
Author Response
Reviewer 2,
We would like to thank you for review of this manuscript. Reviewer suggestions helped us to correct our article. We hope our article after revision will be published in Molecules journal. We provide answers to reviewers comments below. All changes have been added in the text.
Open Review
(x) I would not like to sign my review report
( ) I would like to sign my review report
English language and style
( ) Extensive editing of English language and style required
( ) Moderate English changes required
( ) English language and style are fine/minor spell check required
(x) I don't feel qualified to judge about the English language and style
Yes |
Can be improved |
Must be improved |
Not applicable |
|
Does the introduction provide sufficient background and include all relevant references? |
(x) |
( ) |
( ) |
( ) |
Is the research design appropriate? |
(x) |
( ) |
( ) |
( ) |
Are the methods adequately described? |
( ) |
(x) |
( ) |
( ) |
Are the results clearly presented? |
( ) |
(x) |
( ) |
( ) |
Are the conclusions supported by the results? |
(x) |
( ) |
( ) |
( ) |
Comments and Suggestions for Authors
The problem of heavy metal and metalloid contamination of soil to be a serious barrier to colonization for many plant species is very important. Therefore the effect of the addition of biochar, bentonite, chicken manure, organo-zeolitic substrate to the contaminated copper soil on germination and early growth of the grass species - Agrostis capillaris, A. stolonifera, Festuca rubra and Poa pratensis) was investigated. To this aim, the standard germination indexes, morphometry and biomass analysis were determined. In my opinion, the paper is interesting and deserves to be published after major revision.
Below I have listed some specific comments to improve the paper:
Reviewer: Please finish the sentence ‘effectively reducing the mobility of…’
Authors: Thank you for the warning, we have added to the text mistakenly omitted elements As, Cd, Cu, Ni, Pb, Zn.
Reviewer: In 4.3. Short characteristics of sorbents - some repetitions should be removed.
Authors: It has been done.
Reviewer: What was the reason to use CaCO3? Please add the characteristic of CaCO3.
Authors: We added to the manuscript: “CaCO3 increases the pH and reduce the content of some potentially toxic elements (Pb, Zn, Cd, As), but does not affect the growth of aboveground plant biomass [Tlustoš et al. 2006; Zhong et al. 2015].”
Tlustoš, P.; Száková, K.; Kořinek, K.; Pavlíková, D.; Hanač, A.; Balík, J. The effect of liming on cadmium, lead, and zinc uptakereduction by spring wheat grown in contaminated soil. Plant, Soil and Environ. 2006, 52(1), 16–24.
Zhong, Q.Y.; Zeng, M.; Liao, B.H.; Li, J.F.; Kong, X.Y. Effects of CaCO3 addition on uptake of heavy metals and arsenic in paddy fields. Acta Ecologica Sinica 2015, 35(4), 1242–1248.
Reviewer: As for zeolites the description ‘They are characterized by a well-developed specific surface, high sorption capacity, high ion exchange capacity and resistance to acids and high temperature’ is too general. A detailed zeolite characteristic is needed.
Authors: Thank you for this note. We think that this sentence is not so important (it is really only general information), because we better described the concretely type of zeolite used in experiment and it is perlite. We added the information: “It is added to the soil to prevent its compaction (thus providing aeration and optimal soil moisture retention). It also supports the rooting of plants.”
Reviewer: What does it mean ‘Its size did not need to be adjusted’?
Authors: The sentence has been deleted because we added information that the perlite was in the form of a fine powder.
Reviewer: Please explain why the grass seeds germinated better on the soil contaminated with copper?
Authors: The positive effect of copper on seed germination could probably result from the low copper content in the soil used here. Other experiments show that, as in the case of allelopathic phenomena in plants, small concentrations of chemical compounds can stimulate plant growth, and large concentrations exhibit toxic properties. Examples of this type of research are provided below.
Reviewer: In Conclusion, significant advantages of the proposed solution should be mentioned.
Authors: Conclusion has been corrected.
Sincerely Yours,
Authors

Round 2
Reviewer 2 Report
In my opinion, the paper was revised and deserves to be published.